# Low versus high dose anticoagulation in patients with Coronavirus 2019 pneumonia at the time of admission to critical care units: A multicenter retrospective cohort study in the Beaumont healthcare system

**Kadhim Al-Banaa**[1]*, **Abbas Alshami**[2], **Eiman Elhouderi**[3], **Sally Hannoodee**[4], **Maryam Hannoodee**[5], **Alsadiq Al-Hillan**[6], **Hussam Alhasson**[7], **Faisal Musa**[1], **Joseph Varon**[8,9], **Sharon Einav**[10,11]

1 Department of Hematology and Oncology, Beaumont Hospital, Royal Oak, MI, United States of America, 2 Department of Medicine, Jersey Shore University Medical Center, Neptune, NJ, United States of America, 3 Department of Medicine, Beaumont Hospital, Dearborn, MI, United States of America, 4 Department of Medicine, Einstein Medical Center/Montgomery, Norristown, PA, United States of America, 5 Department of Medicine, Health Quest Medical Practice, Poughkeepsie, NY, United States of America, 6 Department of Gastroenterology, Beaumont Hospital, Royal Oak, MI, United States of America, 7 Department of Medicine, Rochester Regional Health, Rochester, NY, United States of America, 8 Department of Acute and Continuing Care, The University of Texas Health Science Center at Houston, Houston, TX, United States of America, 9 Critical Care Services, United Memorial Medical Center, Houston, TX, United States of America, 10 Intensive Care Unit, Shaare Zedek Medical Centre, Jerusalem, Israel, 11 Faculty of Medicine, Hebrew University, Jerusalem, Israel

* Kadhim.al-banaa@beaumont.org

## Abstract

### Purpose

Coagulopathy is common in patients with COVID-19. The ideal approach to anticoagulation remains under debate. There is a significant variability in existing protocols for anticoagulation, and these are mostly based on sporadic reports, small studies, and expert opinion.

### Materials and methods

This multicenter retrospective cohort study evaluated the association between anticoagulation dose and inpatient mortality among critically ill COVID-19 patients admitted to the intensive care units (ICUs) or step-down units (SDUs) of eight Beaumont Healthcare hospitals in Michigan, USA from March 10th to April 15th, 2020.

### Results

Included were 578 patients with a median age of 64 years; among whom, 57.8% were males. Most patients (n = 447, 77.3%) received high dose and one in four (n = 131, 22.7%) received low dose anticoagulation. Overall mortality rate was 41.9% (n = 242). After adjusting for potential confounders (age, sex, race, BMI, ferritin level at hospital admission, intubation, comorbidities, mSOFA, and Padua score), administration of high anticoagulation

**Data Availability Statement:** All relevant data are within the paper and its Supporting Information files.

**Funding:** The authors received no specific funding for this work.

**Competing interests:** The authors have declared that no competing interests exist.

doses at the time of ICU/SDU admission was associated with decreased inpatient mortality (OR 0.564, 95% CI 0.333–0.953, p = 0.032) compared to low dose.

## Conclusion

Treatment with high dose anticoagulation at the time of ICU/SDU admission was associated with decreased adjusted mortality among critically ill adult patients with COVID-19.

## 1 Introduction

Patients with COVID-19 commonly present with coagulation profiles suggestive of a pro-thrombotic state with elevated fibrinogen [1,2] and, in approximately 60% of severe COVID-19 patients, high D-dimer levels as well [3]. It has also been hypothesized that direct viral effects on the vascular and hemostatic system cause this prothrombotic state [4]. Several studies have reported high rates of venous thromboembolism (VTE) in patients with severe COVID-19 [5,6]. Higher levels of fibrinogen and d-dimers have been shown to correlate with worse outcomes in these patients, including mortality [1,2]. Yet the ideal approach to anticoagulation in patients with COVID-19 remains under debate [7].

Several leading national and international health care institutions have developed protocols for management of thrombotic and antithrombotic therapy related to COVID-19 [8–11]. These protocols are mostly based on sporadic reports and small retrospective studies as large prospective cohorts or interventional studies are still unavailable [12–15]. Based on the assumption that bedridden COVID-19 patients carry similar risk for VTE as do all intensive care patients, the dose of thromboprophylaxis recommended for patients with severe COVID-19 is usually similar to that currently recommended for other critically ill patients [8,9,16].

However, contrary to most critically ill patients, VTE often occurs in COVID-19 patients despite the use of thromboprophylaxis [5,12]. This phenomenon has caused some to support the notion of higher-than-usual doses of anticoagulation drugs especially in patients perceived as being at higher risk for thromboembolism due to remarkably high d-dimer values and/or the presence of additional comorbidities [12,17].

At the time this study was initiated, three large trial platforms were collaborating to assess the benefit of therapeutic versus prophylactic dosing of anticoagulants: The Randomized Embedded, Multi-factorial Adaptive Platform Trial for Community-Acquired Pneumonia (REMAP-CAP), the Accelerating COVID-19 Therapeutic Interventions and Vaccines-4 (ACTIV-4) and the Antithrombotic Therapy to Ameliorate Complications of COVID-19 (ATTACC). The latter has recently paused enrolment of critically ill patients already requiring ICU level of care due to safety concerns shown on interim data analysis, allowing continued enrolment only of patients who are moderately ill.

There remains a significant variability in administration of anticoagulation; dosing is often determined by local trends and experience and is given to the judgment of the physicians at hand. We therefore aimed to investigate the relation between anticoagulation dose and inpatient mortality in critically ill COVID-19 patients. We hypothesized that treatment with high dose anticoagulation decreases inpatient mortality in critically ill COVID-19 patients.

## 2 Materials and methods

The study was conducted in accordance with the REporting of studies Conducted using Observational Routinely-collected health Data (RECORD) Statement. Beaumont Health Institutional Review Board (IRB) reviewed the study protocol and waived the requirement of

informed consent due to the retrospective nature of the study that involves only information collection and analysis involving the investigator's use of identifiable health information when that use is regulated under 45 CFR parts 160 and 164, subparts A and E [HIPAA] (protocol record number 2020–219), and the study protocol was registered in ClinicalTrials.gov database (CT.gov Identifier: NCT04829552).

## 2.1 Study design and setting

This multi-center, retrospective, study was conducted from March 10[th] to April 15[th], 2020. Beaumont Health System (BHS) is the largest healthcare system in terms of inpatient admissions in Michigan (United States). The BHS database is comprised of patient data collected in real time at eight hospitals (overall 3,375 beds). The hospitals are mostly tertiary referral centers, and all provide intensive care or step-down care services. All BHS hospitals are also academic, being affiliated with one of three medical schools (Oakland University William Beaumont School of Medicine, Michigan State University College of Osteopathic Medicine, or Wayne State University School of Medicine) and combined host a total of 93 residency and fellowship training programs.

During the first pandemic wave, COVID-19 patients admitted to BHS intensive care units (ICUs) and step-down units (SDUs) received anticoagulation unless contraindicated, as do other critically ill patients. Standardized dosing was initiated after the end of the first pandemic wave, therefore at the time of data collection for this study the dose of anticoagulation administered depended on individual clinician judgment.

## 2.2 Participants

All adult patients admitted to BHS hospitals with COVID-19 were screened. We used an internal search code based on ICD-10 diagnoses to automatically search BHS electronic medical records for ICU/SDU patients who fulfilled the following inclusion criteria: Age 18 years or older, a positive reverse transcription polymerase chain reaction (rRT-PCR) test for the qualitative detection of nucleic acid from SARS-CoV-2 in upper and lower respiratory specimens as well as peak d-dimer levels exceeding 1,000 mcg/mL and respiratory failure at any time during admission. Respiratory failure was defined as receipt of respiratory support using a high flow oxygen delivery device, non-invasive mechanical ventilation or invasive mechanical ventilation to maintain SaO2 >90% or PaO2 >65 mmHg.

Exclusion criteria were a hospital length of stay less than 5 days, hemorrhage before ICU/SDU admission as this may have precluded/changed management of anticoagulation, treatment with an anticoagulant other than LMWH or unfractionated heparin and constant treatment with the same dose of anticoagulant for less than 5 days.

The control group was comprised of patients treated with subcutaneous LMWH 40 mg once daily or unfractionated heparin 5000 IU twice or three times daily (low dose group). The study group included patients treated with subcutaneous enoxaparin 1 mg/kg twice daily or 1.5 mg/kg daily or a continuous intravenous infusion of unfractionated heparin, with aPTT monitoring every 8 hours to achieve target range of 51–86 seconds per institutional protocol, for at least 5 days (high dose group).

## 2.3 Outcomes

The primary outcome of the study was inpatient all-cause mortality. Secondary outcomes were the rates of major bleeding events (as defined by the International Society on Thrombosis and Hemostasis [18]) and venous thromboses (i.e. deep vein thrombosis, pulmonary embolism). We also planned to study mortality rates in the subgroups of patients with high versus low d-

dimer levels and in the subgroups of intubated versus non-intubated patients receiving study or control treatment.

## 2.4 Variables and data sources

All study data were collected from the electronic medical records (EMRs) of the patients. Beaumont EMR is Epic (Epic Systems Corporation), and all BHS hospitals are interconnected to the same EMR system. Clinicians working in BHS hospitals are similarly instructed, trained, and overseen with regards to documentation. The data were extracted from the Clarity database of Epic EMR system and stored in SharePoint (Microsoft Corporation, Redmond, WA). Data collected included patient demographic variables (i.e., medical record number [MRN], age, sex, race) and medical background diseases (i.e., hypertension, diabetes, respiratory failure, chronic pulmonary disease, common cardiovascular diseases, cancer). We also extracted information regarding conditions other than COVID-19 that may potentially be accompanied by hypercoagulability both congenital and acquired thrombophilia (i.e., factor V Leiden, prothrombin G20210A mutation, protein C and S deficiency, antithrombin deficiency, hepatic failure, rheumatological disorders, smoking and finally, history of VTE). In addition, we collected the medical information required to calculate the modified sequential organ failure assessment (mSOFA) [19] of the patients, Padua scores and d-dimer level at the time of ICU/ SDU admission [20], additional physiological and laboratory characteristics (e.g. body mass index [BMI], ferritin and fibrinogen levels on admission to hospital) as well as the anticoagulation used (type, dose) (Table 1). Data on mortality and the occurrence of major bleeding events (as defined by the International Society on Thrombosis and Hemostasis [18]) and of venous thromboses (i.e. deep vein thrombosis, pulmonary embolism) were also extracted.

## 2.5 Address of bias (missingness)

We examined the missing data for all variables. If the proportion of missing data in any variable exceeded 5%, we compared the characteristics of cases with and without missing data. If data were missing completely at random (MCAR), pairwise deletion was used during analysis. Otherwise, missing at random (MAR) was assumed and multiple imputation was utilized to impute the missing data. Imputation procedures were iterated a minimum of 10 times or until sufficient convergence was achieved.

## 2.6 Sample size calculation

The mortality rates of patients with COVID-19 varies widely, and studies conducted in hospitals with settings similar to ours (combined ICUs/SDUs) had reported a mortality rates ranging between 17% to 30% [21–24]. Therefore, we assumed a mortality rate of 20–25%. Preliminary data review had revealed that the proportion of patients receiving higher doses would be approximately three times higher than that of patients receiving lower doses. Given this proportion, in order to achieve study power of 0.8 for detecting a 10% mortality difference between the groups, we calculated a required sample size of 370–480 high dose patients and 125–160 low dose patients.

## 2.7 Management of quantitative variables

Based on EMR data regarding the type and dose of antithrombotic drugs they had received, patients were classified to either the high or the low dose group.

D-dimer, fibrinogen, and ferritin levels were binned into multiples of upper normal limits to reduce the risk of random chance. Upper normal limits were defined as 500 ng/mL, 300 ng/

**Table 1. Baseline characteristics of participants.**

|  | Median (IQR) or N (%) | | | |
|---|---|---|---|---|
|  | All patients (n = 578) | High dose group (n = 447) | Low dose group n = 131 | P Value |
| Age (years) | 64 (54–76) | 65 (56–74) | 64 (54–76) | 0.89 |
| Sex Male Female | 334 (57.8) 244 (42.2) | 272 (60.9) 175 (39.1) | 62 (47.3) 69 (52.7) | 0.006 |
| Race White Black Asian Indian or Alaskan Other | 205 (35.5) 323 (55.9) 11 (1.9) 2 (3) 37 (6.4) | 156 (34.9) 248 (55.5) 7 (1.6) 1 (0.2) 35 (7.8) | 49 (37.4) 75 (57.3) 4 (3.1) 1 (0.8) 2 (1.5) | 0.073 |
| BMI | 31.6 (27.1–37.4) | 32 (27.3–38) | 31 (26.5–37) | 0.257 |
| No. of Comorbidities | 2 (0–3) | 2 (0–3) | 1 (0–3) | 0.195 |
| Diabetes Mellitus | 217 (37.5) | 164 (36.7) | 53 (40.5) | 0.433 |
| Asthma | 46 (8) | 37 (8.3) | 9 (6.9) | 0.601 |
| Hypertension | 348 (60.2) | 275 (61.5) | 73 (55.7) | 0.233 |
| Atrial Fibrillation | 18 (3.1) | 16 (3.6) | 2 (1.5) | 0.234 |
| Heart Failure | 42 (7.3) | 35 (7.8) | 7 (5.3) | 0.335 |
| COPD | 47 (8.1) | 34 (7.6) | 13 (9.9) | 0.393 |
| Cancer | 56 (9.7) | 46 (10.3) | 10 (7.6) | 0.366 |
| CAD | 80 (13.8) | 66 (14.8) | 14 (10.7) | 0.235 |
| TIA/Stroke | 43 (7.4) | 37 (8.3) | 6 (4.6) | 0.156 |
| CKD | 74 (12.8) | 59 (13.2) | 15 (11.5) | 0.598 |
| VTE (DVT/PE) | 41 (7.1) | 35 (7.8) | 6 (4.6) | 0.203 |
| Other Comorbidities* | 22 (3.8) | 15 (3.4) | 7 (5.3) | 0.296 |
| Ferritin on Admission | 1,195 (531–2,608) | 1,195 (577–2,564) | 1,191 (412–2,817) | 0.693 |
| D-Dimer on ICU Admission | 1,549 (910–3,527) | 1,653 (887–3,614) | 1,444 (1,065–2,281) | 0.313 |
| Fibrinogen on Hospital Admission | 609 (470–752) | 605 (470–741) | 616 (472–732) | 0.833 |
| Intubated | 460 (79.6) | 387 (86.6) | 73 (55.7) | <0.001 |
| mSOFA** | 4 (1–5) | 4 (1–5) | 3 (0–5) | 0.003 |
| Padua score*** | 6 (6–7) | 6 (6–7) | 6 (6–7) | 0.340 |
| Length of stay in ICU or SDU | 20 (13–29) | 22 (14–32) | 13 (9–21) | <0.001 |
| Use of Antiplatelets | 254 (43.9) | 204 (45.6) | 50 (38.2) | 0.129 |
| Duration of intubation (days) | 10 (5–18) | 10 (4–17) | 11 (6–20) | 0.222 |

*Other Comorbidities include: Peripheral arterial diseases, chronic rheumatological conditions, hyperthyroidism, hypothyroidism, and pulmonary hypertension.

**MSOFA score (1.Respiratory:PaO2/FiO2, mmHg, 2. Coagulation: Platelets x103/μL, 3. Liver: Bilirubin, mg/dL, 4. Cardiovascular: Hypotension, 5. CNS, Glasgow: Coma Score, 6. Renal: Creatinine mg/dL, urine output mL/d) [19].

*** Padua score (1. Active cancer, 2. Previous VTE, 3. Reduced mobility, 4. Already known thrombophilia condition, 5. Recent (1 month) trauma and/or surgery, 6. Elderly age (> 65 years). 7. Heart and/or respiratory failure, 8. Acute myocardial infarction or ischemic stroke, 9. Acute infection and/or rheumatologic disorder, 10. Obesity (BMI > = 30), 11. Ongoing hormonal treatment) [20].

mL, and 400 mg/dL for d-dimer, ferritin, and fibrinogen respectively [25,26]. The normal upper limit of ferritin differs according to age and gender but was considered 300 ng/mL for this study [27].

For subgroup analysis we classified the patients into two groups based on their d-dimer levels sampled at the time of ICU/SDU admission (≥2,500 ng/mL or <2,500 ng/mL). This threshold was selected based on prior reports showing mortality benefit with anticoagulation in patients with D-dimer levels above these levels [28].

## 2.8 Statistical analysis

We first used descriptive statistics, frequencies and percentages, to describe categorical variables. We used medians with their interquartile ranges for describing continuous data as these had a non-normal distribution (Shapiro-Wilk test). The Chi square test was used to compare categorical variables, and the Mann-Whitney U test was used to compare continuous variables.

Baseline patient characteristics differed in the control and study groups due to non-random assignment. To reduce the mortality bias created by patient preselection for administration of a high vs. low dose, we used propensity score analysis. The variables identified by univariate analyses as being associated with administration of a high dose (threshold p<0.1) were older age, "other" race, male gender, presence of intubation, and higher mSOFA score. These variables were included in a multivariable logistic regression model to estimate predicted probabilities, which were then used to calculate the inverse probability propensity score weights for administration of a high dose.

To examine the association between the dose of anticoagulation and patient outcomes while accounting for data inflation implied by weighting, robust estimator generalized estimating equation (GEE) binary logistic regression main effect models were created. The models were weighted using the propensity score weights and included variables that had been associated with mortality in univariate analyses (threshold p<0.1). As prior studies had shown interactions between BMI, gender, age and mortality [29], we added BMI and gender and interactions between BMI and age and BMI and gender to the model. We first created correlation matrices and examined these to detect potential multicollinearity and then used the smallest Independence Model Criterion (QIC) to choose between the correlation structures in the different sets of model terms. As shown in Table 2, the final model included overall 18 variables (including interactions), an allowable number for our sample size.

**Table 2. Unadjusted and adjusted parameter estimates of weighted generalized equation estimation to predict inpatient mortality.**

|  | Crude OR |  | p Value | Adjusted OR | 95% CI for OR | p Value |
|---|---|---|---|---|---|---|
| Age | 1.046 | 1.026–1.067 | <0.001 | 1.177 | 1.080–1.272 | <0.001 |
| Female Sex | 0.716 | 0.452–1.135 | 0.155 | 0.176 | 0.025–1.221 | 0.079 |
| Race–White | 1 |  | Ref | 1 |  | Ref |
| Race–Black | 0.627 | 0.391–1.005 | 0.052 | 0.551 | 0.321–0.946 | 0.031 |
| Race–Asian | 0.833 | 0.217–3.194 | 0.790 | 2.116 | 0.237–18.861 | 0.502 |
| Race–Other | 0.290 | 0.089–0.938 | 0.039 | 0.447 | 0.131–1.520 | 0.197 |
| BMI | 1.002 | 0.975–1.029 | 0.900 | 1.229 | 1.061–1.424 | 0.006 |
| BMI*age | - | - | - | 0.997 | 0.995–0.999 | 0.007 |
| BMI*gender (female) | - | - | - | 1.033 | 0.978–1.090 | 0.243 |
| Number of Comorbidities | 1.175 | 1.024–1.347 | 0.021 | 1.136 | 0.976–1.322 | 0.099 |
| D-Dimer (Binned) on ICU/SDU admission | 1.001 | 0.941–1.065 | 0.969 | - | - | - |
| Fibrinogen (Binned) at hospital admission | 0.975 | 0.708–1.344 | 0.879 | - | - | - |
| Ferritin (Binned) at hospital admission | 1.030 | 0.997–1.063 | 0.072 | 1.037 | 1.010–1.065 | 0.007 |
| Intubation | 6.341 | 3.518–11.430 | <0.001 | 9.421 | 4.803–18.482 | <0.001 |
| mSOFA | 1.141 | 1.042–1.250 | 0.005 | 1.043 | 0.941–1.156 | 0.421 |
| Padua score | 1.266 | 1.094–1.465 | 0.002 | 1.062 | 0.901–1.251 | 0.473 |
| Therapeutic Anticoagulation | 0.686 | 0.434–1.085 | 0.107 | 0.564 | 0.333–0.953 | 0.032 |
| Bleeding Event | 0.539 | 0.220–1.321 | 0.176 | - | - | - |
| Thrombosis Event | 0.444 | 0.040–4.965 | 0.510 | - | - | - |

* Race Indian or Alaskan didn't have enough cases and was omitted from the table.

Finally, survival was analyzed using the Cox proportional hazard regression model.

**Subgroup and sensitivity analyses.** As noted above we planned subgroup analysis of inpatient mortality in relation to the treatment protocol based on plasma d-dimer levels before initiation of anticoagulation treatment ($\geq$2,500 ng/mL vs. <2,500 ng/mL). We also preplanned subgroup analysis for intubated versus non-intubated patients.

We conducted a sensitivity analysis for the outcome by using an alternative cutoff (21-day mortality instead of inpatient mortality). We conducted a second sensitivity analysis for the statistical approach by using logistic regression model only to adjust for covariates instead of propensity score weighting to account for imbalances and regression model for residual differences [30].

All analyses were performed using IBM SPSS Statistics version 25.0 (IBM Corporation, Artmonk, NY). Unless otherwise specified, an alpha value (p) of 0.05 was used to determine statistical significance.

## 2.9 Data access and cleaning methods

Three of the authors (KA, EE, FM) were given full access to the Beaumont database for the purpose of data retrieval. The data were deidentified after retrieval and transfer to SPSS. Data cleaning procedures included removal of illogical values if present (e.g. age greater than 150 years, multiple digit responses when only single digit responses were possible, dates that have not yet occurred), removal of duplicate values, correction of typos where possible and removal of data where the response with a typo was unclear. Data conversion was not required as all Beaumont data are unified. Data cleaning was performed by the authors with full database access.

## 3 Results

During the study period overall 704 patients were admitted to Beaumont Health ICUs/SDUs with COVID-19 and among these 578 fulfilled inclusion criteria (Fig 1).

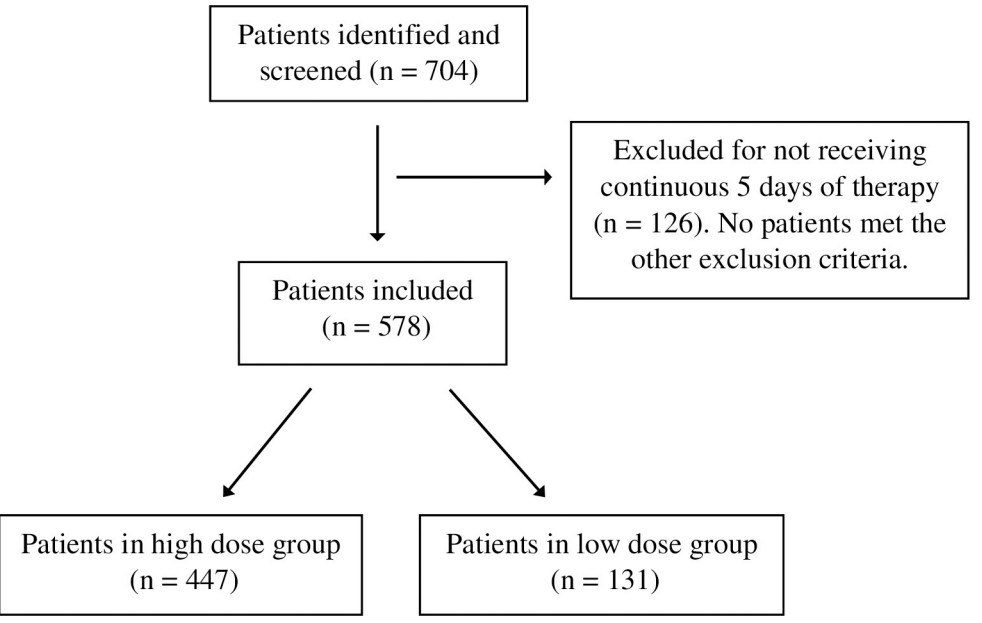

**Fig 1. Inclusion and exclusion flow chart.**

The cohort included overall 57.8% males (n = 334) with a median age of 64 years (IQR, 54–76). Their median ICU/SDU length of stay was 20 days (IQR, 13–29). The overall proportion of patients that underwent mechanical ventilation in this ICU/SDU cohort was 79.6% (n = 460). The median length of mechanical ventilation was 10 days (IQR = 5–18). The overall proportion of patients with a bleeding event was 4.7% (n = 27). The overall mortality rate was 41.9% (n = 242), and the mortality rate among those undergoing mechanical ventilation was 48.9% (n = 225).

## 3.1 Missing data

We identified more than 5% missing values in d-dimer levels at the time of ICU/SDU admission and in fibrinogen levels at the time of hospital admission [16.3% (n = 94) and 15.9% (n = 92) respectively].

Missing data on D-dimer levels at the time of ICU/SDU admission was statistically associated with race, fibrinogen level, the dose of anticoagulation administered and the occurrence of a thrombotic event. Missing data on D-dimer levels at the time of ICU/SDU admission was not associated with age, gender, number of comorbidities, BMI, ferritin levels, patient mSOFA scores, Padua scores, the occurrence of a bleeding event, intubation status, duration of intubation, length of ICU/SDU stay or mortality.

Missing data on fibrinogen levels at the time of hospital admission was statistically associated with patient BMI, D-dimer levels at the time of ICU/SDU admission, the dose of anticoagulation administered, the length of ICU/SDU stay, intubation status and duration of intubation. No association was found between missing data on fibrinogen levels and age, gender, race, number of comorbidities, ferritin levels at the time of hospital admission, patient mSOFA scores, Padua scores, the occurrence of a thrombotic or bleeding event, or mortality.

We therefore assumed that missing occurred at random (MAR) but not completely at random (MCAR). We used a fully conditional specification method of multiple imputation using weighted linear regression model for variable scaling to create 5 imputations with 10 iterations. This provided sufficient convergence when plotted against means and standard deviations for both of the variables with missing values (D-dimer on admission to ICU/SDU and fibrinogen level on admission to hospital).

## 3.2 Comparison of control and study group baseline characteristics

Overall, 77.3% (n = 447) of the patients received a high anticoagulant dose and 22.7% (n = 131) received a low anticoagulant dose. The control and study groups were similar in age, number and composition of co-morbidities and body mass index. The groups were also well matched in terms of ferritin and fibrinogen levels at the time of hospital admission, Padua scores and d-dimer levels at the time of ICU/SDU admission.

Among patients receiving a high dose of anticoagulation there were more male patients (60.9%, n = 272) whereas among those receiving a low dose gender representation was almost equal (47.3% males and 52.7% females, n = 69). The two groups also differed in mSOFA scores which were higher among those receiving a high dose of anticoagulation (Table 1).

## 3.3 Unadjusted and adjusted comparison of control and study group mortality

Unadjusted inpatient mortality rates were similar with high (43%, n = 192) and low (38.2%, n = 50) doses of anticoagulation (p = 0.329).

The variables associated with inpatient mortality (p<0.1) in univariate analyses were older age, white and Asian races (versus black and other races), a greater number of comorbidities,

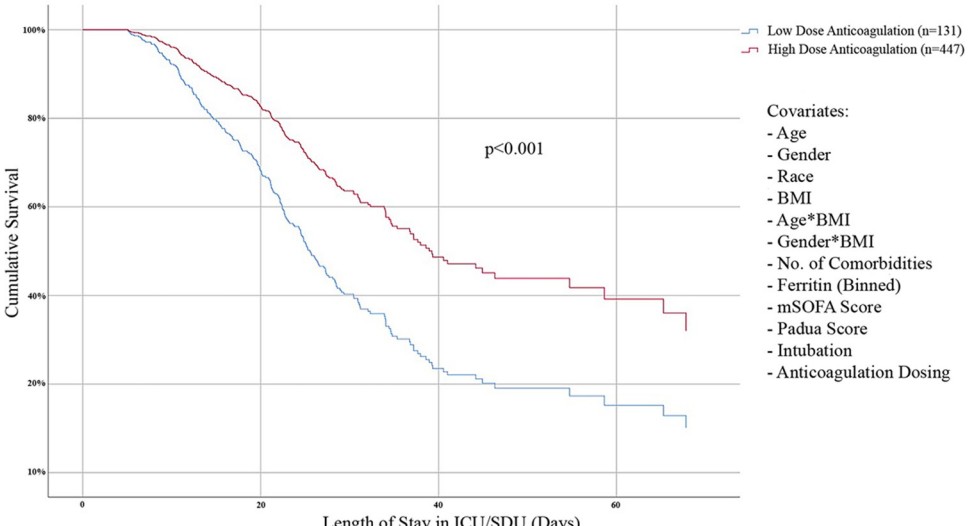

**Fig 2. Cox proportional hazard cumulative survival (all cause inpatient mortality) for critically ill patients receiving high versus low dose anticoagulation after adjusting for the listed confounders.**

higher ferritin levels at the time of hospital admission, higher mSOFA score, higher Padua score and intubation. We adjusted for these potential confounders, as well as BMI, gender and interactions between BMI and age and BMI and gender. In the weighted GEE model, administration of a high dose of anticoagulant was independently associated with lower inpatient mortality (OR 0.564, 95% CI 0.333–0.953, p = 0.032) (Table 2). Sensitivity analyses revealed consistent results. The association of high dose anticoagulation with 21-day mortality was comparable with that of inpatient mortality (adjusted OR 0.507, 95% CI 0.292–0.881, p = 0.016). The association of high dose anticoagulation using binary logistic regression model to adjust for covariates (without propensity score weighting) was also comparable (adjusted OR 0.594, 95% CI 0.356–0.992, p = 0.046). Cox proportional hazard regression model also showed decreased inpatient mortality with therapeutic anticoagulation (adjusted OR 0.5, 95% CI 0.355–0.698, p<0.001) (Fig 2).

### 3.4 Subgroup analyses

Subgroup analysis of patients with d-dimer levels ≥ 2500 ng/mL at the time of ICU/SDU admission (n = 183) showed that high dose anticoagulation was associated with lower inpatient mortality (adjusted OR 0.413, 95% CI 0.155–1.105) but this finding did not achieve statistical significance (p = 0.07).

Subgroup analysis by intubation status showed that high anticoagulation doses were associated with lower inpatient mortality among patients that underwent intubation (adjusted OR 0.523, 95% CI 0.295–0.925, p = 0.03) but not among those that did not undergo intubation (adjusted OR 0.477, 95% CI 0.092–2.478, p = 0.379).

### 3.5 Incidence of adverse events

Bleeding occurred in 4.7% of patients (n = 27). The rate of major bleeding events was 1.6% (n = 7) with high versus 0.8% (n = 1) with low dose anticoagulation (p = 0.45). The unadjusted rate of any bleeding event did not differ with high (5.1%, n = 23) versus low (3.1%, n = 4) dose

anticoagulation (p = 0.32). No patient who received high dose, versus 2.3% (n = 3) of the patients who received low dose anticoagulation, had VTE (p = 0.001).

## 4 Discussion

This multicenter cohort study included more than 500 patients admitted with critical COVID-19 respiratory failure to the ICUs/SDUs of eight hospitals. We classified and compared patients who received low anticoagulant doses described as "prophylactic" to those receiving high anticoagulant doses usually described as "therapeutic." Patients receiving high doses were more often male and had higher mSOFA scores. Unadjusted inpatient mortality rates were not associated with the dose of anticoagulant received and approximated 40% in both groups. However, after adjustment for other characteristics associated with mortality (i.e., older age, gender, BMI, race, number of comorbidities, and disease severity), a high anticoagulant dose was independently associated with lower inpatient mortality. This association was maintained in sensitivity analyses, was also observed in Cox regression analysis, and seemed even stronger in patients that underwent intubation.

The mortality rate among our cohort is higher than that described in prior studies of the association between anticoagulation and mortality [13,31], but mortality rates similar to ours have been described in other critically ill COVID-19 cohorts [32,33]. Our results are consistent with those of Paranjpe et al., who retrospectively studied mortality in relation to anticoagulation among 2,773 patients hospitalized with laboratory-confirmed COVID-19 in the Mount Sinai Health System in New York City. Their Cox proportional hazards model adjusted for demographic characteristics and comorbidities but not for acute disease severity as did ours. Paranjpe et al. did not preselect critically ill patients for the analysis. However, among their subset of patients who required mechanical ventilation (n = 395), in-hospital mortality was lower (29.1% vs 62.7%) and median survival (21 days vs. 9 days) was longer with high dose versus low dose anticoagulation [13]. Lynn et al. contradicted these findings, showing a survival advantage among patients receiving therapeutic versus prophylactic anticoagulation doses, which disappeared in the subset of critically ill patients requiring ICU admission [31]. However, these authors also noted that intubated patients who received therapeutic anticoagulation had a slightly higher survival probability in the first four days of treatment [31]. As their mean duration of therapeutic anticoagulation was somewhat longer than that of the Mt. Sinai cohort, they proposed that the survival benefit shown by Paranjbpe et al. may have stemmed from timing bias. We preselected inpatient survival as our primary outcome and performed a sensitivity analysis on the timing of the outcome, making such bias unlikely. Our cohort was also comprised entirely of critically ill patients, three fourths of which underwent intubation and mechanical ventilation. In our cohort, therapeutic anticoagulation was started at the time of ICU/SDU admission. This information was not provided in other studies and may not have been the case in other critically ill patient cohorts.

Recently released two reports on three large, open-label, adaptive, multiplatform, randomized clinical trials (REMAP-CAP, ACTIV-4, and ATTACC) [34,35]. These three trials have harmonized protocols and a composite primary outcome, an ordinal scale that combined in-hospital mortality and days free of organ support to day 21 [34,35]. The first report included an analysis of severely ill patients with Covid-19 requiring ICU level of care who were randomized within the first 48 hours of admission to the ICU [35]. Randomization of patients was stopped when the prespecified criterion for futility was met, and a total of 1,098 patients were included in the final analysis (534 received therapeutic anticoagulation and 564 received usual care pharmacological thromboprophylaxis). Unlike our results, authors demonstrated that although therapeutic dosing did not result in a greater probability of survival to hospital

discharge or a greater number of days free of cardiovascular or respiratory organ support than did usual care prophylactic dosing despite fewer thrombotic events noted in the therapeutic dosing group (6.4% vs. 10.4%) [35]. Notably, many participants in the usual care arm received an intermediate dose of thromboprophylaxis due to the change in national practice guidelines in the United Kingdom for patients admitted to the ICU where the majority of enrollment happened. In addition, the enrollment of patients was stopped due to futility. Halting rules for clinical trials require complex data interpretation. Data is also often required from prior studies to understand whether the minimum magnitude of treatment benefit is large enough to offset treatment harms. This is still not the case with critically ill COVID-19 patients. Prudence also demands that less evidence be required to stop the trial for harm than for benefit.

Moreover, our population demographically differed from their population (56% of our population is Black, compared to ~5% in their study), potentially contributing to the notable difference in treatment effects. The second report of the three trials mentioned above included an analysis of the moderately ill Covid-19 patients who required hospitalization but not ICU level of care [34]. They enrolled 2,219 patients in the final analysis; 1,181 patients received therapeutic dosing, and 1,050 patients received prophylactic dosing. The therapeutic dosing improved the survival to hospital discharge without organ support compared to prophylactic dosing (80.2% vs. 76.4%) [34]. The difference in the treatment effect noted between the two reports suggests that the timing of administration of therapeutic dose anticoagulation plays a crucial role in treatment effects.

The results of another open-label, multicenter, randomized, controlled trial have also been published recently [36]. In this trial, hospitalized patients with COVID-19 infection and elevated D-dimer were randomized in a 1:1 fashion to receive treatment with therapeutic dose anticoagulation with rivaroxaban for clinically stable patients or heparin products for clinically unstable patients or treatment with standard in-hospital prophylactic anticoagulation. No difference in primary outcomes, which was defined as a hierarchical analysis of time to death, duration of hospitalization, or duration of supplemental oxygen to day 30, was found between the two groups (34.8% versus 41.3% (win ratio 0.86 [95% CI 0.59–1.22], p = 0.40). In addition, an increased risk of major or clinically relevant non-major bleeding was found in the therapeutic group (8% versus 2% with a relative risk 3.64 [95% CI 1.61–8.27], p = 0.0010). However, the majority of patients (94%) were clinically stable, and only a small fraction of them (6%) were clinically unstable. Another explanation for the lack of effect of rivaroxaban in the ACTION study, compared with heparin and its derivatives in our study, is the possible pleiotropic effects of heparin which inhibit multiple coagulation proteases, might have other anti-inflammatory and antiviral effects [37]. Also, the difference in the oral route of rivaroxaban administration may have affected the findings as hospitalized patients might have abnormal absorption of oral anticoagulation, leading to erratic and variable effects. Finally, Lung microvascular thrombosis contributing to respiratory worsening in COVID-19 might not be primarily preventable by factor Xa inhibition but possibly mainly by direct thrombin inhibition or antithrombin activation.

Whether d-dimer levels can be used to identify high-risk patients who should receive a larger dose of anticoagulation remains controversial. Tang et al. studied patients described as having severe COVID-19 (n = 449) and noted that mortality increased as d-dimer levels increased among patients who did not receive prophylactic anticoagulation. No such effect was seen among those receiving prophylactic anticoagulation [16]. Rising d-dimer levels have been proposed to reflect worsening endovascular disease, and anticoagulation has been proposed to protect against this destructive effect of SARS-Cov-2 on endothelial cells [4]. Retrospective studies have shown higher mortality rates among COVID-19 patients with higher d-dimer [38–40] and fibrinogen levels [38]. This has led several authors to propose that

anticoagulant dosing should be adjusted according to D-dimer levels. McBane et al. suggested a cutoff level of 3.0 ug/mL (3,000 ng/ml), reflecting a 6-fold increase above the upper limit of normal [41]. Tassiopoulos et al. suggested an escalating thromboprophylaxis protocol based on daily D-dimer levels [42]. In our cohort, the dose of anticoagulant administered did not modify the association between d-dimer levels and outcome, but our study was not powered for this analysis.

Major bleeding (as per ISTH) occurred in 4.7% of patients (n = 27) in our study. Al-Samkari et al. reported a bleeding rate of 7.6% overall and 5.6% major bleeding in ICU patients with COVID-19 receiving prophylactic dose anticoagulation [43]. Llitjos et al. reported bleeding rates of 3.4% among all COVID-19 patients and 7.3% among patients receiving high-dose heparin in medical wards [12]. Pesavento et al. retrospectively compared COVID-19 patients treated with prophylactic doses versus higher doses of anticoagulants (n = 240 vs. n = 84) and showed somewhat higher rates of major bleeding and death among patients who received higher doses of anticoagulants. However, these authors specifically excluded critically ill patients from their study [44]. An online pre-publication by Gonzalez-Porras et al. describes an unselected adult patient population (n = 611) retrospectively stratified according to three LMWH regimens with the lowest mortality but more major hemorrhage with high dose anticoagulation [15].

The incidence of VTE was low in our study compared to others [5,44]. This low rate may reflect under-ascertainment. BHS ICUs/SDUs do not routinely screen critically ill COVID-19 patients for VTE. Access to duplex testing was restricted as part of the infection containment measures implemented in our hospitals. Clinicians were possibly disinclined to transport unstable patients to diagnostic computed tomography angiography, and a diagnosis of VTE would not have changed management for critically ill patients already receiving therapeutic anticoagulation (i.e., such testing was clinically redundant). Finally, we sought no other manifestations of hypercoagulability. This is an important limitation given that there have been reports of in-situ thrombosis, albeit mainly pulmonary, in patients with severe COVID-19 [45].

Our study has several additional limitations. Retrospective studies are more likely to be biased by confounding. Our study and control groups were relatively well balanced, and we adjusted for potential confounders that were identified. However, there may be additional, unidentified confounders that led to selection bias. We also did not control for concomitant therapies. This is often a study limitation in patients with COVID-19 [46]. Like others who have studied this topic in COVID-19 patients, we have no laboratory evidence of anticoagulant effects; heparin infusion was directed by aPTT, but the dose of LMWH was not titrated to AntiXa levels. In order to make inappropriate dosing less likely, we excluded patients receiving inconsistent doses and those not within the dosing range usually classified as "prophylactic" and "therapeutic."

## 5 Conclusion

In this retrospective multicenter study of critically ill adults with COVID-19, early administration of anticoagulation in doses classified as therapeutic was associated with reduced adjusted inpatient mortality rates when compared to doses classified as prophylactic. Although the overall bleeding rate was higher among patients receiving higher doses, the rate of major bleeding complications remained relatively low in both groups and more VTEs were seen with lower anticoagulation doses. Additional data is required to clarify how the dosing and the timing of anticoagulation affect the outcomes of critically ill adults with COVID-19.

## Supporting information

**S1 Data. Data coding.**
(DOCX)

**S2 Data. De-identified data.**
(SAV)

## Acknowledgments

We would like to thank Dr. Richard F. W. Barnes, PhD for his critical revision of the statistical analyses.

## Author Contributions

**Conceptualization:** Kadhim Al-Banaa, Faisal Musa.

**Data curation:** Kadhim Al-Banaa, Sally Hannoodee, Maryam Hannoodee.

**Formal analysis:** Abbas Alshami.

**Investigation:** Kadhim Al-Banaa.

**Methodology:** Kadhim Al-Banaa, Abbas Alshami.

**Supervision:** Joseph Varon, Sharon Einav.

**Writing – original draft:** Kadhim Al-Banaa.

**Writing – review & editing:** Kadhim Al-Banaa, Abbas Alshami, Eiman Elhouderi, Alsadiq Al-Hillan, Hussam Alhasson, Sharon Einav.

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
