## [Decision Letter · Decision Letter 0]

30 Dec 2021

PONE-D-21-34752Low versus high dose anticoagulation in patients with Coronavirus 2019 pneumonia at the time of admission to critical care units: A multicenter retrospective cohort study in the Beaumont Healthcare SystemPLOS ONE

Dear Dr. Al-Banaa,

Thank you for submitting your manuscript to PLOS ONE. After careful consideration, we feel that it has merit but does not fully meet PLOS ONE’s publication criteria as it currently stands. Therefore, we invite you to submit a revised version of the manuscript that addresses the points raised during the review process.

We look forward to receiving your revised manuscript.

Kind regards,

Tai-Heng Chen, M.D.

Academic Editor

PLOS ONE

Journal Requirements:

Reviewers' comments:

Reviewer's Responses to Questions

**Comments to the Author**

1. Is the manuscript technically sound, and do the data support the conclusions?

Reviewer #1: Yes

Reviewer #2: Yes

2. Has the statistical analysis been performed appropriately and rigorously? 

Reviewer #1: Yes

Reviewer #2: Yes

3. Have the authors made all data underlying the findings in their manuscript fully available?

Reviewer #1: Yes

Reviewer #2: Yes

4. Is the manuscript presented in an intelligible fashion and written in standard English?

Reviewer #1: Yes

Reviewer #2: Yes

5. Review Comments to the Author

Reviewer #1: The authors conducted a study of treatment with high dose anticoagulation at the time of ICU/SDU admission which was

associated with decreased adjusted mortality among critically ill adult patients with COVID-19 infection. The major concern of this retrospective study is how to decide the dosage of anticoagulation. Since the D-dimer and fibrinogen levels were similar between the two groups, how the doctors used higher or lower dosage of anticoagulation?

Reviewer #2: At the outset,a big thank you for allowing me to review this article.It is a very interesting study and like all other studies on COVID,there are a lot of grey areas.I have some questions for the authors.

1)How many of these patients were on vasopressors,CRRT,how many required NIV support and how many were on high flow 02 ,although a mention was made about the intubated patients.Was there any correlation between these and mortality.

2)Was there any association between x ray pattern,CT severity score and D dimer levels.

3)Was a bed side 2 D echo done and if so how many of them had PAH or RV dysfunction.

4)Was there sepsis induced coagulopathy. Were procalcitonon levels done in this study ?

Possible association between development of antiphospholipid antibodies ,notably lupus anticoagulant (LAC) has been seen in many studies which may contribute to coagulopathy,was there any such observation in this study?

5) How many of these patients received steroids,remdesevir?There has been a weak association between the treatment modalities used in COVID especially more so with corticosteroids resulting in a thrombogenic state.

6. PLOS authors have the option to publish the peer review history of their article (what does this mean?). If published, this will include your full peer review and any attached files.

Reviewer #1: **Yes: **HSIEH MING-HSIUNG

Reviewer #2: No

---

## [Author Response · Author response to Decision Letter 0]

18 Feb 2022

Review Comments to the Author

Reviewer #1:

Comment: The authors conducted a study of treatment with high dose anticoagulation at the time of ICU/SDU admission which wasassociated with decreased adjusted mortality among critically ill adult patients with COVID-19 infection. The major concern of this retrospective study is how to decide the dosage of anticoagulation. Since the D-dimer and fibrinogen levels were similar between the two groups, how the doctors used higher or lower dosage of anticoagulation?

Answer: Thank you for highlighting this important question. This study utilized retrospective data at the beginning of the pandemic. There was only little known about the effect of anticoagulation on Covid-19 outcomes. Therefore, thedecision of anticoagulation dosage was based entirely on the treating physician preference in a near random fashion. Indeed, this was one of the points that encouraged us to conduct this study, as less bias would be anticipated. The manuscript is updated to reflect this point.

Reviewer #2: 

Comment: How many of these patients were on vasopressors,CRRT,how many required NIV support and how many were on high flow 02,although a mention was made about the intubated patients.Was there any correlation between these and mortality?

Response: We agree with the reviewer on the potential effect of these factors on the investigated outcomes. These variables were accounted for through the mSOFA score, as a composite measure of end organ damage. We did not include these variables as individual variables. The admission criteria to our ICU/SDU units at that time included patients on high flow, NIV, or intubated, and divided the population into intubated and non-intubated (both High flow and NIV populations). It was felt that the sample size will be very small for these populations (NIV and High flow patients individually) and any conclusions drawn on these populations will be under powered, therefore they were included as one group.

Comment: Was there any association between x ray pattern,CT severity score and D dimer levels?

Answer: We did not study the imaging findings for these patients. D-dimer level was investigated, but there was no association between its level and inpatient mortality, thus it was not included in the multivariable analysis. In addition, we performed a subgroup analysis based on the D-dimer level.

Comment: Was a bed side 2 D echo done and if so how many of them had PAH or RV dysfunction?

Answer: Unfortunately, we did not collect the echocardiography studies results for these patients. We do not believe a significant number of patients had echocardiography done at that time due to lack of PPE, unless absolutely necessary.

Comment: Was there sepsis induced coagulopathy. Were procalcitonin levels done in this study?Possible association between development of antiphospholipid antibodies,notably lupus anticoagulant (LAC) has been seen in many studies which may contribute to coagulopathy,was there any such observation in this study?

Answer: Coagulopathy, as part of mSOFA score, was included in the study but not a separate variable. No data were collected regarding procalcitonin or lupus anticoagulant was investigated in the study.

Comment: How many of these patients received steroids,remdesivir?There has been a weak association between the treatment modalities used in COVID especially more so with corticosteroids resulting in a thrombogenic state.

Answer: Thank you for highlighting this important potential interaction. During the time of conducting this study, neither the Remdesivir nor the steroids were approved for treatment of Covid-19, therefore none of our patient were treated with them and we can’t explore such interactions. The manuscript was updated to reflect these limitations.

---

## [Decision Letter · Decision Letter 1]

11 Mar 2022

Low versus high dose anticoagulation in patients with Coronavirus 2019 pneumonia at the time of admission to critical care units: A multicenter retrospective cohort study in the Beaumont Healthcare System

PONE-D-21-34752R1

Dear Dr. Al-Banaa,

We’re pleased to inform you that your manuscript has been judged scientifically suitable for publication and will be formally accepted for publication once it meets all outstanding technical requirements.

Kind regards,

Tai-Heng Chen, M.D.

Academic Editor

PLOS ONE

Reviewers' comments:

Reviewer's Responses to Questions

**Comments to the Author**

1. If the authors have adequately addressed your comments raised in a previous round of review and you feel that this manuscript is now acceptable for publication, you may indicate that here to bypass the “Comments to the Author” section, enter your conflict of interest statement in the “Confidential to Editor” section, and submit your "Accept" recommendation.

Reviewer #1: All comments have been addressed

Reviewer #2: All comments have been addressed

2. Is the manuscript technically sound, and do the data support the conclusions?

Reviewer #1: Yes

Reviewer #2: Yes

3. Has the statistical analysis been performed appropriately and rigorously? 

Reviewer #1: Yes

Reviewer #2: Yes

4. Have the authors made all data underlying the findings in their manuscript fully available?

Reviewer #1: Yes

Reviewer #2: Yes

5. Is the manuscript presented in an intelligible fashion and written in standard English?

Reviewer #1: Yes

Reviewer #2: Yes

6. Review Comments to the Author

Reviewer #1: no more comments

The authors have good response to my previous comments and also have good answers to other reviewers' comments.

Reviewer #2: Bypass,all comments have been addressed and there are no conflicts of interests,the article is recommended for publication

7. PLOS authors have the option to publish the peer review history of their article (what does this mean?). If published, this will include your full peer review and any attached files.

Reviewer #1: **Yes: **Ming-Hsiung Hsieh

Reviewer #2: No

---

## [Editor Report · Acceptance letter]

16 Mar 2022

PONE-D-21-34752R1 

Low versus high dose anticoagulation in patients with Coronavirus 2019 pneumonia at the time of admission to critical care units: A multicenter retrospective cohort study in the Beaumont Healthcare System 

Dear Dr. Al-Banaa:

I'm pleased to inform you that your manuscript has been deemed suitable for publication in PLOS ONE. Congratulations! Your manuscript is now with our production department. 

Kind regards, 

on behalf of

Dr. Tai-Heng Chen 

Academic Editor

PLOS ONE